# Liquid Biopsy: A Game Changer for Type 2 Diabetes

**DOI:** 10.3390/ijms25052661

**Published:** 2024-02-25

**Authors:** Gratiela Gradisteanu Pircalabioru, Madalina Musat, Viviana Elian, Ciprian Iliescu

**Affiliations:** 1eBio-Hub Research-Center, National University of Science and Technology “Politehnica” Bucharest, 6 Iuliu Maniu Bulevard, Campus Building, 061344 Bucharest, Romania or gratiela.gradisteanu@icub.unibuc.ro (G.G.P.); ciprian.iliescu@upb.ro (C.I.); 2Research Institute of University of Bucharest, University of Bucharest, 050095 Bucharest, Romania; 3Academy of Romanian Scientists, 3 Ilfov Str., 050094 Bucharest, Romania; 4Department of Endocrinology, Carol Davila University of Medicine and Pharmacy, 030167 Bucharest, Romania; 5Department of Endocrinology, C.I. Parhon National Institute of Endocrinology, 011683 Bucharest, Romania; 6Department of Diabetes, Nutrition and Metabolic Diseases, Carol Davila University of Medicine and Pharmacy, 5-7 Ion Movila Street, 030167 Bucharest, Romania; 7Department of Diabetes, Nutrition and Metabolic Diseases, Prof. Dr. N. C. Paulescu National Institute of Diabetes, Nutrition and Metabolic Diseases, 030167 Bucharest, Romania; 8National Research and Development Institute in Microtechnologies—IMT Bucharest, 126A Erou Iancu Nicolae Street, 077190 Voluntari, Romania

**Keywords:** type 2 diabetes, liquid biopsy, exosomes, miRNA, molecular diagnostic

## Abstract

As the burden of type 2 diabetes (T2D) continues to escalate globally, there is a growing need for novel, less-invasive biomarkers capable of early diabetes detection and monitoring of disease progression. Liquid biopsy, recognized for its minimally invasive nature, is increasingly being applied beyond oncology, and nevertheless shows its potential when the collection of the tissue biopsy is not possible. This diagnostic approach involves utilizing liquid biopsy markers such as cell-free nucleic acids, extracellular vesicles, and diverse metabolites for the molecular diagnosis of T2D and its related complications. In this context, we thoroughly examine recent developments in T2D liquid biopsy research. Additionally, we discuss the primary challenges and future prospects of employing liquid biopsy in the management of T2D. Prognosis, diagnosis and monitoring of T2D through liquid biopsy could be a game-changing technique for personalized diabetes management.

## 1. Introduction

The rapid increase in prevalence of type 2 diabetes mellitus (T2D) qualifies it as one of the fastest-growing global health emergencies of the 21st century [1]. T2D is a chronic condition necessitating continuous management and attention to avert its extensive health consequences, including myocardial infarction, cardiac failure, blindness, kidney failure, stroke, and amputation.

T2D is a multifactorial disease resulting from a combination of factors, including inadequate beta-pancreatic cell (β-cell) function, reduced insulin sensitivity, and chronic inflammation that precede the diabetes onset with up to 15 years. Currently, the diagnosis of T2D is made typically once hyperglycemia has become evident, a limitation that impedes early disease detection and precludes the possibility of timely interventions aimed at preserving β-cell mass and improving patient outcomes. Liquid biopsy can be a promising solution to this challenge, being useful in the diagnostic, management, and prognostic of the disease and providing a deeper understanding of the disease’s pathophysiology.

Early detection and intervention are crucial for preventing the progression of T2D and reducing the risk of complications such as nephropathy, cardiovascular disease, neuropathy, or retinopathy. Even though biomarkers such as glycemia or HbA1c are widely used for diagnosing T2D, presently, there is no biomarker that is able to predict the ongoing destruction of beta cells in the pancreas in the early stages of the disease as opposed to T1D, where specific autoantibodies against pancreatic islet cells or insulin are often present before the onset of symptoms and individuals at increased risk or in the early stages of T1D can be timely identified.

On the other hand, liquid biopsy shows its potential in the prognosis, diagnosis, and monitoring of cancer [2,3,4]. The use of a small amount of “body fluids” (usually blood), containing biomarkers that can provide relevant information for diagnostic and monitoring, present some relevant advantages such as reduced invasiveness, easy protocol for sample collection, lower costs, and real-time information [5,6,7].

This review study was conducted at the eBio-Hub Research Center affiliated to the National University of Science and Technology Politehnica Bucharest. PubMed, Scopus, Google Scholar, and Web of Science were searched using the following keywords: “type 2 diabetes”, “biomarker”, “microRNA”, “liquid biopsy”, and “cell free DNA.”

The main objectives of this review study are as follows: (i) to conduct a critical review of recent advancements in the field of T2D liquid biopsy and assess the current state of research and technology in T2D liquid biopsy; (ii) to provide an overview of the primary challenges associated with T2D liquid biopsy and explore the perspectives surrounding the potential of liquid biopsy to revolutionize personalized T2D management; (iii) to highlight the potential role of liquid biopsy as a game changer in advancing the field of personalized T2D management; and (iv) to address any gaps or limitations in current research on T2D liquid biopsy and to propose potential avenues for future research and development to enhance the applicability and effectiveness of liquid biopsy in T2D management.

## 2. Liquid Biopsy Markers in T2D

Biomarkers that allow for early diagnosis and effective monitoring have the potential to lead to better patient-centric therapeutic decisions. In T2D, identifying the precise timing of critical events such as β-cell stress, de-differentiation, or death is essential for developing effective treatments [8]. However, the inaccessibility of pancreatic tissue in humans for longitudinal, minimally invasive monitoring represents a significant obstacle. To address this shortcoming, liquid biopsy could be employed for the detection of β-cell markers. Liquid biopsies encompass a diverse range of clinically informative elements. Blood samples can undergo analysis for metabolites, exosomes, and cell-free nucleic acids, each offering valuable numerical and molecular insights. Employment of liquid biopsy in T2D patients can provide valuable insight regarding T2D complications, cardiovascular disease (CVD), and cancer risk. An overview of the key elements and the impact of the liquid biopsy markers in T2D is presented in Figure 1.

### 2.1. Cell-Free Nucleic Acids

#### 2.1.1. Circulating Cell-Free DNA (ccfDNA)

Circulating cell-free DNA (ccfDNA), also known as cell-free DNA (cfDNA), refers to short (≈160 nt) double-stranded DNA fragments found in various biological fluids, including blood, urine, saliva, and cerebrospinal fluid. Generally, ccfDNA is actively released by living cells or can result after cell death (apoptosis, necrosis) [9].

Several methods have been utilized to assess cfDNA, including droplet digital polymerase chain reaction (ddPCR), beads, emulsion, amplification, and magnetics (BEAMing), tagged-amplicon deep sequencing (TAm-Seq), whole-genome bisulfite sequencing (WGBS-Seq), whole-exome sequencing (WES), and whole-genome sequencing (WGS).

BEAMing proves to be a cost-effective and highly sensitive screening method for identifying established mutations. The method of integrating PCR with flow cytometry has the capability to identify alterations at levels as minimal as 0.01%, demonstrating excellent concordance with tissue testing [10].

Droplet digital polymerase chain reaction (ddPCR) is capable of detecting genomic material at levels as low as 0.01–1.0%, making it a valuable tool for identifying potentially rare mutations and quantifying copy number variants [11]. Nevertheless, its applicability is limited to assessing the presence of well-characterized sequences.

In patients with cancer, whole-exome sequencing offers a comprehensive analysis of all existing tumor mutations, allowing for the identification of potential oncogenes and tumor suppressor genes. However, its sensitivity may be comparatively lower than other methods due to the inclusion of exomic alterations. Despite this, it stands out for its cost-effectiveness and high yield [12].

Tagged-amplicon deep sequencing (TAm-Seq) enables a highly specific and sensitive analysis, achieving an accuracy of approximately 97%. It has the capability to detect DNA levels as low as 2% through the utilization of primers to tag and identify genomic sequences. TAm-Seq boasts a high sequencing throughput, leading to reduced sequencing time and cost, allowing for the simultaneous sequencing of millions of DNA molecules. However, it is essential for the desired sequence to be pre-characterized for the methodology to be effective [13].

Whole-genome sequencing (WGS) examines the entirety of the sample genome to identify well-characterized and detrimental alterations, along with variants of unknown significance. It holds significant potential for a thorough assessment of genetic mutations and, in case of T2D, could be used to identify cancer risk; however, its application is constrained by challenges in quality assurance, ethical considerations, time, and cost. The interpretation of results can be challenging outside of specialized centers [12].

Whole-genome bisulfite sequencing (WGBS-Seq) stands as the benchmark in DNA methylation analysis. It provides a measurement for each cytosine with exceptional accuracy. While it has the capability to uncover partially methylated domains in cancer cells, the sensitivity of this method can be compromised due to potential variations in DNA degradation [12].

Cell-free DNA (cfDNA)-based biomarkers have shown promise as minimally invasive options for the early diagnosis and monitoring of diabetes in various studies [14,15].

CcfDNA fragments retain characteristics indicative of their tissue of origin. Determining the ccfDNA tissue of origin can be performed using different approaches: the detection of distinct SNPs and/or genetic mutations, genome-wide methylation or 5hmC profiling, and fragmentation profiling by nucleosome footprints [16].

Adipose tissue dysfunction, characterized by hypertrophy, inflammation, and cell death, is closely associated with the development of insulin resistance—a key feature of T2D. Hence, the detection of cfDNA fragments associated with adipose tissue degeneration can provide insights into the mechanisms underlying insulin resistance [17].

Combining cfDNA concentration with tissue-specific epigenetic signatures may offer a comprehensive approach to assessing metabolic risk and disease progression. For instance, Lehmann-Werman et al. used methylation profiling to identify circulating DNA linked to different pathological processes: pancreatic β-cell DNA from patients with type 1 diabetes and exocrine pancreatic DNA from patients with pancreatic cancer [18].

Karaglani et al. evaluated the methylation profile of a panel of specific genes related to β-cells, including INS (insulin), GCK (glucokinase), IAPP (islet amyloid polypeptide-amylin), KCNJ11 (potassium inwardly rectifying channel subfamily J member 11), and ABCC8 (ATP binding cassette subfamily C member 8) and, based on the data generated, they built classifying predictive models using automated machine learning. *INS*, *IAPP*, *GCK*, and *KCNJ11* levels differed significantly between T2D patients and healthy controls. Based on ccfDNA parameters, methylation data and demographical information, automated machine learning analysis generated biosignatures, including *GCK*, *IAPP,* and *KCNJ11* methylation, with a very high discriminating performance of T2D from healthy individuals [19].

While human leukocyte antigen (HLA) typing is primarily used for assessing the risk of developing type 1 diabetes, there is ongoing research exploring potential associations between specific HLA alleles and the risk of diabetes-related complications. These complications may include diabetic nephropathy, retinopathy, and neuropathy. HLA typing, combined with other genetic and clinical information, may contribute to individualized risk assessments for diabetes complications. Identifying individuals at higher risk may allow for targeted monitoring and early interventions to prevent or manage complications. Ongoing research is exploring the complex interplay between genetics, including HLA alleles, and the development of diabetes and its complications. Advances in genomic medicine and personalized medicine may lead to more refined risk assessments and tailored therapeutic approaches. The influence of variations in the HLA genotype on the development of type 2 diabetes complications is not clearly understood. Recently, Birinci et al. (2022) identified the HLA variant HLA-DR7 to be associated with diabetic rethynopathy development [20]. The presence of the HLA-DQB1*0501 allele demonstrated a protective association against diabetic nephropathy (DN) in individuals of Han ethnicity in China [21].

Cell-free mitochondrial DNA (ccf-mtDNA) is produced and discharged from cells into the systemic circulation in response to cellular damage or stress. This release may occur passively through various forms of cell death or actively from living cells through processes that are not fully comprehended at present [22]. Increased blood ccf-mtDNA levels have been associated with T2D associated with coronary heart disease or cognitive impairment [23,24].

#### 2.1.2. Circulating Cell-Free RNA (ccfRNA)

Human blood is home to a wide range of cfRNA species, including microRNA (miRNA), messenger RNA (mRNA), long-noncoding RNA(lncRNA), Piwi-interactingRNA (piRNA), circular RNA (circRNA), transfer RNA (tRNA), and miscellaneous other noncoding RNA molecules [25]. 

Advances in high-throughput sequencing and bioinformatics have improved the ability to profile and analyze cfRNA. Emerging single-cell RNA sequencing (scRNA-seq) technologies enabled the study of cfRNA heterogeneity at the single-cell level [26]. 

#### 2.1.3. miRNAs

Circulating miRNAs, small 17- to 23-bp-long noncoding RNAs have been proposed as potential biomarkers for T2D [27].

In streptozotocin-treated mice, miRNA-375, which is highly abundant in pancreatic islet cells, was reported to be an early indicator of T1D [28].

miRNA-375, miRNA-101, and miRNA-802, which have roles in pancreatic islet function and insulin secretion, have been identified as being significantly increased in T2D patients [29].

Elevated levels of miR-122 were shown to be strongly associated with an increased risk of developing T2D [30]. MicroRNA-122 levels were also found to be increased in patients with diabetic retinopathy [31].

Large-scale studies are necessary for clinical miRNA validation. Since miRNAs are involved in diverse regulatory processes, future studies are needed to unravel their roles in the overall pathogenesis of T2D. A shift from targeted analysis towards exploratory next generation sequencing (NGS) should be performed in order to potentially discover novel miRNA relevant for T2D pathogenesis [32].

Circulating long-noncoding RNAs (lncRNAs), which are RNA molecules longer than 200 nucleotides and that do not code for proteins, exhibit cell and tissue specificity and have been implicated in various cellular processes, including transcriptional regulation. In the context of T2D complications, circulating lncRNAs have been shown to play important roles as transcriptional activators and regulators. In a mouse model of diabetic nephropathy, Tug1 lncRNA was shown to modulate mitochondrial bioenergetics in glomerular podocytes [33].

Other lncRNAs such as GAS5, RNA-MEG3is, MALAT1, and CYP4B1-PS1-001 were reported to be involved in diabetes-induced microvascular dysfunction and inflammation [34,35,36,37].

The use of lncRNAs comes with several disadvantages, however, including a lack of conservation across species and technically challenging analysis. Also, the fact that lncRNAs harbor a multitude of roles at the cellular level makes their specific functional characterization difficult. 

Even though cfRNA provides a snapshot of transcripts reflecting the health status of multiple RNAs, understanding the precise origins of cfRNA, including their cell types of origin, can be challenging due to the complex and dynamic nature of cfRNA in circulation [38]. 

### 2.2. Extracellular Vesicles

EVs play a significant role in intercellular communication and their shedding is influenced by intracellular elements such as calcium ions. EVs can interact with recipient cells through various mechanisms, including ligand–receptor binding, phagocytosis, direct fusion with plasma membranes, micropinocytosis, and lipid raft-mediated internalization or endocytosis [39].

EVs are classified based on their different sizes and biogenesis processes (Figure 2). The main types of EVs are microparticles (MPs), exosomes, and apoptotic bodies, each with distinct characteristics [40]. MPs that are also known as microvesicles or ectosomes originate from the outward budding or shedding of the plasma membrane of cells and have a diameter size range from 30 nm to approximately 1 μm. Exosomes originate from the inward budding of endocytic vesicles within the cell, forming early endosomes which mature into late endosomes, also known as multivesicular bodies (MVBs), where intraluminal vesicles (ILVs) are formed through the inward budding of the endosomal membrane. Exosomes are released when MVBs fuse with the cell’s plasma membrane, leading to the extracellular release of the ILVs as exosomes. Exosomes are generally smaller, typically ranging from 30 to 150 nm in diameter. Apoptotic bodies are distinct from MPs and exosomes in terms of their origin and are typically larger than MPs and exosomes [41].

#### 2.2.1. Exosomes

Exosomes play a role in cell-to-cell communication by carrying proteins, lipids, RNA, and other molecules between cells. Exosomes have been implicated in various physiological and pathological processes, including those associated with diabetes [42]. The role of exosomes in diabetes-related pathophysiology, including vascular complications, inflammation, and coagulation changes, is an area of growing research interest. 

While exosomes play a crucial role in the early detection and treatment of various conditions [43], their small size, low density (1.13–1.19 g/mL), and coexistence with similar components such as cell fragments and proteins in bodily fluids present significant challenges for their isolation [44]. Standardized methods for separation and quantification are crucial for both the research and clinical use of exosomes. In order to develop more efficient and rational exosome separation technologies, it is essential for us to comprehend the current separation technology landscape. Common separation techniques include ultracentrifugation, ultrafiltration, polymer precipitation, immunoaffinity, and size exclusion chromatography. However, these methods yield low isolation efficiency, sample loss, and compromised exosome recovery and purity. Ultracentrifugation, commonly regarded as the “gold standard” for exosome isolation, involves expensive instruments, large sample volumes, potential protein contamination, and complex isolation steps [45].

Proteins in exosomes obtained from the body fluids of individuals with T2D exhibit variations. For instance, dipeptidyl peptidase-IV (DPP IV), known for inactivating glucagon-like peptide-1 (GLP-1), is linked to T2D. Moreover, the microvesicle-bound form of DPP IV is the predominant type found in urine, and its levels are notably elevated in individuals with T2DM compared to control subjects [46]. Patients with diabetes exhibiting proteinuria show markedly elevated levels of Wilms tumor protein 1 in urinary exosomes, suggesting that exosomes can serve as early biomarkers for podocyte injury in individuals with diabetes [47].

Patients with T2D and associated microvascular complications exhibit markedly elevated levels of miR-7 in exosomes derived from serum compared to those without such complications. Consequently, alterations in these exosomal biomarkers precede changes at the organ level and offer greater specificity than examining whole urine or blood. This further strengthens the potential role of exosomes in the early diagnosis of diabetes and its complications [48].

Several studies have reported elevated levels of exosomes in the blood samples collected from individuals with diabetes, including both type 1 diabetes (T1D) and T2D. A comparative cross-sectional study performed on normoglycemic individuals and patients with prediabetes or diabetes mellitus reported that diabetics harbored elevated levels of plasma extracellular vesicles and that exosome concentration was positively associated with the insulin resistance index [49]. Meta-analyses have demonstrated a significant rise in circulating exosomes released by platelets, monocytes, and endothelial cells in individuals with diabetes. However, there is no discernible difference in exosomes from leukocytes between patients with diabetes and the control group [50].

The communication between different cell types via exosomes can promote the development of different T2D complications. For instance, endothelial cell-derived exosomes can interact with macrophages and smooth muscle cells altering their behavior and subsequently leading to atherosclerotic plaque formation [51]. 

Atherosclerotic lesions in the aortic endothelium can affect other normal cell types within the vascular wall via the transport of regulatory proteins carried by exosomes. Transfer of exosomes from the blood of db/db diabetic mice into db/m+ non-diabetic mice can severely impair the endothelial cell function in recipient mice [52].

Exosomes could serve as early biomarkers of podocyte injury in diabetic kidney disease (DKD) since significantly elevated levels of Wilms tumor protein 1 (WT1) were found in urinary exosomes of patients with diabetes and proteinuria [47].

It is known that vasa vasorum (VV) angiogenesis is increased for T2D patients and may promote atherosclerotic plaque rupture. Wang et al. [53] demonstrated the role of insulin resistance adipocyte-derived exosomes in modulating vasa vasorum angiogenesis tested on ApoE(−/−) mice.

Goetzl et al. [54] investigated the cargo proteins of human plasma endothelial cell-derived exosomes in patients with atherosclerotic cerebrovascular disease. Platelets exhibited significantly higher levels of platelet glycoprotein VI, integrin-linked kinase-1, high-mobility group box-1 protein, chemokine CXCL4, and thrombospondin-1. 

#### 2.2.2. Microbial Extracellular Vesicles

EVs are essential in various biological processes, including immune response regulation, tissue repair, and cell signaling. Microbiota-derived EVs (MEVs) have gained attention for their role in host cell–microbiota communication [55]. MEVs can transport molecules from bacteria, archaea, fungi, and other microorganisms to host cells, influencing host immune responses and other physiological processes.

EVs derived from gut microbiota have been linked to various host functions, including the regulation of the immune system and the suppression of cancer. There is also a suggestion that these EVs may play a role in modulating the gut–brain axis, as studies have reported their ability to reach the central nervous system and regulate cerebral function [56]. Apart from exhibiting beneficial effects and contributing to host homeostasis, changes in the profile of EVs derived from gut microbiota have been associated with the progression of several diseases, such as HIV, inflammatory bowel disease, or cancer therapy-induced intestinal mucositis.

It has been proposed that EVs derived from gut microbiota, possessing pro-inflammatory properties, might be sufficiently mild to be beneficial. They could contribute to maintaining gut barrier integrity and intestinal homeostasis by inducing host immune and defense responses. However, in cases of dysbiosis or host susceptibility to inflammatory disorders, an imbalance in the abundance of pro-inflammatory EVs from gut microbiota, initially considered beneficial to the host, may potentially result in exacerbated inflammatory responses that are ultimately harmful. Supporting this hypothesis, Hickey et al. observed that EVs derived from the commensal *Bacteroides thetaiotaomicron* could be taken up by intestinal macrophages, triggering an inflammatory response by inducing the production of pro-inflammatory cytokines (such as TNF-α and IL-1β) and contributing to the development of colitis in genetically susceptible mice [57].

There is growing evidence to suggest that the interplay between gut microbiota, their EVs, and the host’s immune system can have significant consequences for metabolic health, including the potential to improve obesity and diabetes. Indeed, some studies have indicated that interventions aimed at modulating the gut microbiota and their EVs, such as dietary changes or the use of probiotics or prebiotics, may lead to improvements in metabolic diseases. 

A study by Chelakkot et al. found that oral administration of *A. muciniphila* EVs to mice that were fed a high-fat diet (HFD) resulted in a reduction in gut barrier permeability, reduced body weight gain, and improved glucose tolerance [58]. The beneficial effects of A. muciniphila EVs were confirmed by Ashrafian et al. who reported that *A. muciniphila* EVs had significant effects on alleviating weight gain and adiposity in HFD-induced obese mice [59]. Conversely, Choi et al. showed that that the stool EVs from HFD-fed mice had an adverse effect on glucose metabolism, promoting insulin resistance in both skeletal muscle and adipose tissue [60]. 

There are current constraints in characterizing EVs derived from gut microbiota, posing challenges in identifying the molecules responsible for their physiological effects. The isolation and characterization of the diverse profile of gut microbiota are considerably more intricate due to the absence of universal markers for bacterial EVs and their size similarity to mammalian EVs, thus complicating their differentiation from mammalian EVs in bodily fluids. Some authors have identified the bacterial origin of fecal EVs by amplifying DNA from the most prevalent phyla of gut microbiota through PCR [61]. However, the comprehensive identification of the origin of biofluids, fecal-derived EVs, and EV-associated molecules like miRNAs remains challenging due to the complex nature of the material.

#### 2.2.3. Metabolites

While diagnosing diabetes or prediabetes currently relies on a straightforward blood glucose measurement, there are compelling reasons to advocate for concurrent metabolite assessments. Firstly, relying solely on a single measurement of plasma glucose for diagnosing diabetes or prediabetes can potentially yield false positive results, leading to misdiagnosis by healthcare professionals. In such scenarios, metabolite indices can serve as valuable complementary indicators. Secondly, these metabolites can offer a wealth of additional information about patients with diabetes or prediabetes.

In determining the progression to prediabetes and diabetes, there are several widely used and standardized biomarkers such as glycemia, HbA1c, glycated albumin, and fructosamine. The need to diagnose T2D even before β-cell function is lost determined the need to find other potential useful biomarkers, and a part of them are detailed in Table 1.

Further research is needed to pinpoint the most precise biomarkers, acknowledging that relying on a singular determinant is likely to have inherent constraints. Hence, combining several biomarkers could provide a more precise forecast of individuals with an increased likelihood of progressing from prediabetes to diabetes.

Several metabolite signatures linked to T2D and its progression have been recently described, and these include an alanine-to-glycine ratio, acylcarnitines, fetuin A, and branched chain amino acids [79,80,81,82,83]. However, these metabolites are still undergoing investigation and necessitate future validation in larger cohorts. Upon identification and validation in clinical studies, these biomarkers could potentially be integrated into point-of-care devices or wearable sensors, facilitating their rapid detection.

## 3. Salivary Markers

The conventional monitoring of serological parameters in diabetes typically involves invasive techniques, causing discomfort and distress. Collecting saliva is convenient, enhancing accessibility for both healthcare professionals and patients and is especially beneficial in scenarios where obtaining blood samples may pose challenges or discomfort. Saliva comprises a range of biomarkers, encompassing DNA, RNA, proteins (3000 proteins and 12,000 peptides), and metabolites [84]. An elegant review by Srinivasan et al. presented the wide array of salivary biomarkers linked to T2D, including glucose, glycosylated hemoglobin (HbA1c), amylase, alpha 2-macroglobulin, alpha defensins, carbonyls, dehydroprogesterone, ghrelin, high-density lipoprotein, C reactive protein, leptin, insulin, Il-1β, osteopontin, osteocalcin, transferrin receptor, urea, and uric acid [85,86].

In contrast to genetic T2D biomarkers, there is a limited body of research on mRNA biomarkers derived from saliva samples. A study examined differential gene expression in saliva from 13 T2D patients, revealing elevated levels of KRAS, SAT1, SLC13A2, and TMEM72, alongside reduced expressions of EGFR and PSMB2 genes [87]. Initially identified through microarray analysis, these six biomarkers were subsequently validated in a cohort of 13 T2D patients and 13 healthy controls. A logistic model incorporating four of these salivary biomarkers (KRAS, SAT1, EGFR, and PSMB2) successfully distinguished T2D patients from healthy controls, yielding an area under the curve (AUC) of 0.917 with a 95% confidence interval of 0.809–1.000. This suggests that salivary mRNAs hold promise as T2D biomarkers, although further investigations involving larger sample sizes are warranted. Subsequently, a validated panel comprising four salivary extracellular RNA biomarkers (IL1R2—interleukin 1 receptor type 2, VPS4B—vacuolar protein sorting 4 homolog B, CAP1—cyclase-associated actin cytoskeleton regulatory protein 1, LUZP6—leucine zipper protein 6) along with body mass index (BMI) has been identified. This panel demonstrates the capability to differentiate individuals with high and low insulin resistance, both in the general population and within subgroups of participants who are healthy and pre-diabetic [88].

There is still a need to identify reliable salivary biomarkers in order to diagnose prediabetes or to distinguish different T2D complications. For instance, differentiating between diabetic nephropathy (DN) and nondiabetic renal disease (NDRD) is crucial for tailoring more specific treatments. However, currently, there are no ideal biomarkers for making this distinction.

A recent study by Han et al. (2022) screened different salivary glycopatterns in DN and NDRD patients using lectin microarrays, with validation conducted through lectin blotting. Diagnostic models were then constructed using logistic regression and artificial neural network analyses, and their validity was confirmed in another cohort [89].

Oxidative stress occurs early in the development of T2D, often preceding the onset of clinical symptoms. Thus, monitoring oxidative stress markers allows for the detection of subtle changes at a molecular level before overt metabolic disturbances become evident [90]. Oxidative stress and inflammation can affect pancreatic beta cells, reducing their function and insulin secretion, further exacerbating glucose dysregulation. Prolonged exposure to oxidative stress, inflammation, and protein glycation contributes to the development of diabetic complications, including cardiovascular disease, neuropathy, nephropathy, and retinopathy. Understanding and targeting these interconnected pathways is essential for developing therapeutic interventions to manage T2D and prevent associated complications [91].

Numerous studies suggest that an imbalance in oxidant/antioxidant mediators plays a significant role in the development and advancement of metabolic syndrome, T2D, and cardiovascular disease. However, the majority of research has concentrated on the distribution of these indicators in tissues and blood, leaving their influence on saliva composition less explored. Products resulting from lipid peroxidation, protein oxidation, and DNA damage can be directly evaluated in saliva, potentially providing a means to diagnose systemic disorders associated with oxidative stress. The evaluation of salivary redox biomarkers appears to be applicable for both diagnosing and monitoring various health conditions, including obesity, diabetes, hypertensive disorders, and heart failure. In these conditions, there is a pathological depletion of molecules and enzymes with antioxidant properties in saliva, while oxidative and nitrosative by-products are favored. For instance, research has demonstrated elevated salivary oxidative biomarkers, including 4-hydroxynonenal (4-HNE), 8-isoprostanes (8-isoP), advanced oxidation protein products (AOPP), protein carbonyl groups (PC), and 8-hydroxy-D-guanosine (8-OHdG) [92].

The use of saliva samples for the molecular diagnostic of T2D comes with several caveats. For example, the local oral status and pathologies related to the oral cavity, such as periodontitis and dental caries may influence the redox balance of saliva, posing potential interference with its widespread routine clinical application [93]. Some biomarkers, such as pro-inflammatory cytokines, are common to many diseases, whereas others are more specific. The presence of other inflammatory comorbidities could also influence the reliability of salivary markers. Moreover, the oral microbiota, specific to each individual, might produce different metabolites and other signaling molecules which might hinder an accurate diagnostic.

The potential to identify a specific array of candidate molecules using saliva for distinguishing each stage of chronic disorder and creating panels of salivary mediators as appropriate molecular biomarkers, when integrated with the demographic, genetic, and anthropometric characteristics of patients, could offer an innovative diagnostic tool. Nevertheless, there is still a need to identify salivary markers specific for different T2D stages and complications. Once identified and validated in clinical studies, these biomarkers could further be incorporated into point-of-care devices or wearable sensors to enable their fast detection.

## 4. Challenges and Perspectives

The evident clinical validity of liquid biopsy is underscored by numerous ongoing or completed clinical trials seeking to demonstrate that decisions based on circulating components could improve diagnostic and enhance patient survival. Despite promising data from observational studies, albeit with limited patient numbers, the full clinical utility of liquid biopsies is yet to be conclusively established. Detecting and characterizing circulating biomarkers in the early disease stages pose challenges due to their low concentrations in patient biofluids. In oncology, advancements in circulating tumor DNA (ctDNA) analysis, utilizing ultra-sensitive state-of-the-art next-generation sequencing technologies and patient-specific panels, have facilitated tumor characterization, assessment of responses to neoadjuvant chemotherapies, and detection of minimal residual disease (MRD) post-surgery. While this methodological revolution has given rise to various NGS procedures with a shared goal of detecting ultra-low diluted cfDNA, important considerations for clinical extrapolation, particularly in terms of technical complexity, still remain. This complexity is especially pronounced at the bioinformatic level, where the development of non-specialized, user-friendly software is crucial for broader clinical application.

An important aspect to consider is the fact that genetic differences such as variations in allele frequencies, mutation patterns, or genomic rearrangements among racial and ethnic groups can impact the composition of cfDNA in liquid biopsies [94].

Hormonal factors, such as estrogen and testosterone levels, may influence the biology of certain cancers. Liquid biopsy results might reflect these hormonal influences, impacting the interpretation of biomarkers in male and female patients.

Discovering signatures for noncoding RNA in the blood is more demanding compared to other commonly used biomarkers. This is attributed to their low abundance in biofluids, the absence of suitable housekeeping noncoding RNA reference analytes, and elevated intra-patient variability.

Exosomes can carry bioactive molecules that contribute to inflammation and oxidative stress in blood vessels, ultimately leading to vascular damage. These elevated levels may be associated with diabetes-related complications and may serve as potential biomarkers for the disease. However, over the last few decades, a major limitation in clinical applications has been the drawbacks associated with conventional exosome isolation methods. This challenge has prompted efforts to create emerging separation platforms. Integration of emerging technologies such as microfluidics, electrical, centrifugal, and acoustical forces into exosome isolation technologies has advanced significantly and become more sophisticated. Microfluidic devices tailored for exosome isolation and purification are anticipated to be promising tools for early detection and biomedical applications [43].

Despite the notable progress, it is evident that current separation methods are not without their flaws. Understanding the role of exosomes in diabetes-related pathophysiology may open up new avenues for therapeutic interventions. Recent studies explore the use of exosomes for targeted drug delivery and regenerative medicine approaches. For example, mesenchymal stem cell exosomes contain growth factors and cytokines that can stimulate cell proliferation and tissue repair, promoting the regeneration of T2D damaged tissues [95].

Microfluidics-based techniques for the isolation of EVs [43] have gained significant attention due to their unique advantages, including the ability to combine EV isolation and disease detection on a single platform. Microfluidic-based systems harbor many advantages such as portability, small starting volume, fast isolation, and cost-efficiency. EVs can be separated from smaller cellular debris, proteins, and other particles using microfluidic nanowire and micropillar structures [96].

Future research should focus on understanding the biogenesis, cargo, and functions of EVs as it is crucial for unravelling their roles in cellular communication and their impact on health and disease. In addition to achieving high-purity exosome isolation, the development of more integrated, high-throughput, and high-recovery-rate devices will open up promising avenues for advances in exosome-based diagnostics and biomedical applications in the years to come. In addition, liquid biopsy combined with machine learning and AI will enable the practice of precision medicine by identifying the specific molecular characteristics of T2D patients to decipher the risk for complication development as well as helping in selecting targeted therapies that are most likely to be effective for a particular patient, minimizing unnecessary side effects.

In routine clinical practice, variations in individual responses to treatment may arise, and reliable indicators predicting glycemic response and the likelihood of adverse events in a given patient are currently lacking. The need to identify markers relevant for the patient drug response should be taken into account in the field of molecular diagnostic. For example, SGLT2 inhibitors, a novel category of antihyperglycemic medications, operate within the proximal tubules of the kidney. Various mutations within the SLC5A2 (solute carrier family 5 member 2) gene, which encode SGLT2, have been associated with familial renal glucosuria, characterized by unusually elevated urinary glucose excretion even in the presence of normal blood glucose levels. These mutations can affect SGLT2 expression, membrane localization, or transporter function. Alongside rare missense mutations, common genetic variants in the SLC5A2 gene have been documented, potentially influencing glucose homeostasis and playing a role in the risk of developing T2D as well as affecting the response to SGLT2 inhibitor treatments [97].

Another target for molecular detection in T2D patients could be represented by peroxisome proliferator-activated receptors (PPARs), which are fatty acid-activated transcription factors of nuclear hormone receptor superfamily that are well-established regulators of lipid metabolism, mitochondrial biogenesis, and energy homeostasis. Their activation has central implications in the function of oxidative tissues and organs such as cardiomyocytes, liver, and muscle. These receptors are important in the context of understanding the molecular mechanisms driving diabetic cardiomyopathy. Drugs such as pemafibrate, metformin, and GLP-1 agonists, classified as PPARα-related drugs, have exhibited effectiveness and safety in clinical studies by reducing lipid and glucose levels in patients with diabetes [98]. Hence, analyzing these receptors as liquid biopsy markers could aid in the personalized treatment of T2D patients at risk of developing cardiomyopathy.

There is an urgent need for more interventional clinical trials to facilitate the widespread adoption of liquid biopsy in clinical practice. For this, the standardization of preanalytical and analytical methods across multiple centers is crucial before liquid biopsy can be reliably incorporated into clinical settings. Policymakers and business leaders should actively engage in these clinical trials and discussions to inform national and international decisions. Existing international consortia (i.e., European Liquid Biopsy Society, BLOODPAC) are already spearheading significant initiatives to accomplish this mission, working towards the development and validation of a comprehensive set of standard operating procedures.

## 5. Conclusions

T2D has significant clinical implications, not only for glucose regulation, but also for overall metabolic and cardiovascular health. Understanding the underlying mechanisms of T2D is essential for developing effective treatments and interventions that address the causes of the disease, beyond managing symptoms.

Currently, there is an imperative need for the development of minimally invasive biomarkers, which aligns with patient-centered care, focusing on early detection, personalized treatment, and improved quality of life for individuals with T2D.

Collaboration across academia, the biotechnology and pharmaceutical industries, and various stakeholders is essential for advancing the field. We foresee that precision medicine strategies, which pinpoint high-risk patient populations and predict specific therapeutic benefits, will play a pivotal role in this progression.

Finally, the cost-effectiveness of traditional diagnostic tools versus innovative liquid biopsy biomarkers can vary depending on several factors, including the specific medical condition being diagnosed, the stage of disease, and the overall healthcare infrastructure. Adoption of innovative diagnostic tools can come with additional costs for the healthcare system. The regulatory landscape and the degree of adoption by healthcare professionals can influence the cost-effectiveness of diagnostic tools. Newly developed technologies may face higher initial costs due to research and development investments. The extent of insurance coverage for different diagnostic methods can play a crucial role in their cost-effectiveness for both healthcare providers and patients.

Hence, the cost-effectiveness of traditional diagnostic tools versus liquid biopsy biomarkers is a complex and dynamic issue. It often involves a trade-off between upfront costs, invasiveness, speed of results, and long-term outcomes. As technology advances and more evidence becomes available, the landscape may shift, and the choice between traditional and innovative diagnostic methods may become more nuanced.

## Figures and Tables

**Figure 1 ijms-25-02661-f001:**
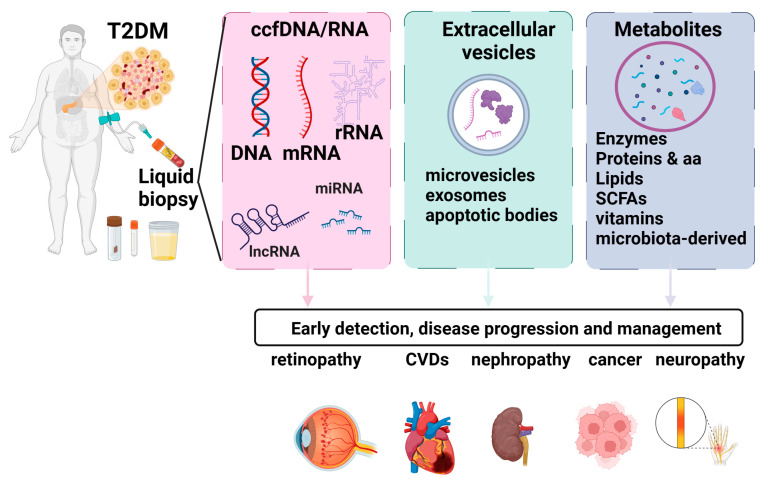
Liquid biopsy markers in T2D. Liquid biopsies encompass a diverse range of clinically informative elements. Blood samples can undergo analysis for metabolites, exosomes, and cell-free nucleic acids, each offering valuable numerical and molecular insights. Employment of liquid biopsy in T2D patients can provide valuable insight regarding T2D complications, cardiovascular disease (CVD), and cancer risk. Figure created using biorender.com.

**Figure 2 ijms-25-02661-f002:**
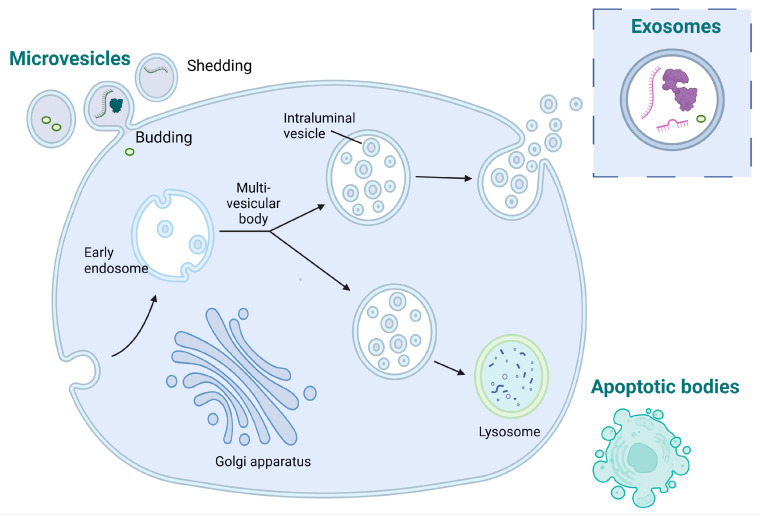
Diagram illustrating the processes involved in the generation of microvesicles, exosomes, and apoptotic bodies.

**Table 1 ijms-25-02661-t001:** Biomarkers linked to detection of prediabetes and diabetes. Abbreviations: T2D—type 2 diabetes, LADA—latent autoimmune diabetes of the adult, CKD—chronic kidney disease, FPG—fasting plasma glucose, IFG—impaired fasting glucose, CVD—cardiovascular disease, SGLT2i—sodium-glucose inked transporter 2 inhibitors, IR—insulin resistance.

Biomarker	Advantages	Disadvantages	Prediabetes Detection	Diabetes Diagnosis
Classical Biomarkers	
Glycated Hemoglobin(HbA1c)	StandardizedReplicable, easy to performReflects chronic hyperglycemia (90 days)Less daily perturbationsCorrelates with the risk of microvascular complications and mortality	Moderate sensitivity as excludes extreme glycemic values and variabilityNot reliable when hemoglobinopathies, anemia or other red blood cell alterations are present	Sensitivity: 0.354Specificity: 0.834 [62]	Sensitivity: 0.59Specificity: 0.96[63,64]
Fasting Plasma Glucose	Cheap, standardized, easy to performCorrelated with insulin secretion impairment and insulin resistance	Varies on daily conditions such as stress, illness, diet, fasting	Sensitivity: 0.43Specificity: 0.94[65]	Sensitivity: 0.82Specificity: 0.89[Kaur] [64]
2 h Plasma Glucose	Cheap, standardizedCorrelated with insulin secretion and insulin resistance	Requires special conditions (fasting, glucose ingestion)	Sensitivity: 0.81[62]	Sensitivity: 0.93[62]
Glycated Albumin	Replicable, easy to performReflects glycemia over 14–21 daysNot sensitive to anemia, Hb-pathies, CKD, pregnancyUseful in therapeutic changes follow-upPredictor of cardiovascular complications and risk of hospitalization or death	Not reliable in nephrotic syndrome, hyperthyroidism, glucocorticoid or iron therapy, malnutrition, and advanced liver diseaseInfluenced by adipose tissueNot standardized for diagnosis of prediabetes	Sensitivity: 0.41Specificity: 0.71[66]	Sensitivity: 0.83Specificity: 0.83[67,68]
Fructosamine	Easy to performNot influenced by mealsReflects glycemia over 14–30 daysUseful in pregnancyPossible correlation with microvascular complications [69]	Not reliable in nephrotic syndrome, hepatic disease, cirrhosis, thyroid disease, and malnutrition	Limited data	Sensitivity: 0.82Specificity: 0.94[70]
1,5-Anhydroglucitol	Reflects glycemia over 7–14 daysUseful in diagnosing diabetes if correlated with FPG [71]Correlates with the risk of CVD [72]	Not reliable in CKD or SGLT2i use	Sensitivity: 0.78Specificity: 0.72 [71]	Sensitivity: 0.96Specificity: 0.88[73]
Adiponectin	Inversely related to prediabetes and type 2 diabetes [74]Good predictor of development towards diabetesCorrelated with insulin sensitivity	Influenced by lifestyle interventions and adipose tissue		
Inflammatory biomarkers	
CRP, IL6	Increase with the progression from IFG to T2DEasy to perform, cheapAssociated with prediabetes, type 2 diabetes and insulin resistance [75]	Significantly influenced by other inflammatory diseases		
Fibrinogen, PAI	Independently associated with insulin levels and prediabetes and type 2 diabetes [76]	Low specificity		
Il18	Correlated with the presence of prediabetes, T2D, and LADA [77], [78]Associated with metabolic syndrome	Expensive, influenced by other inflammatory states		
Promising Biomarkers	
Branched-Chain Amino Acids BCAA [79]	Highly correlated with BMI, insulin levels, HOMA-IR	Influenced by metabolic imbalancesModestly associated with IFG, IGT		
Fetuin-A	Correlated with increased risk of T2D and complications.Independently correlated with T2D [80]Possibly involved in IR	Not enough data		
Alanine-to-Glycine ratio [81]	Can predict incidence and remission of T2DStronger if associated with obesity	Still under investigation		
MicoRNAs [82]	Involved in β-cell function and synthesis and secretion of insulin	Still under investigation		
Acylcarnitines [83]	Involved in insulin resistance	Few data in prediabetes

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
