# Peer review of "Liquid Biopsy: A Game Changer for Type 2 Diabetes"

_ijms, 2024, doi:10.3390/ijms25052661_

Round 1
Reviewer 1 Report
Comments and Suggestions for Authors
New and less intrusive biomarkers that can identify diabetes early and track the disease's progression are increasingly needed as the prevalence of type 2 diabetes (T2D) rises worldwide. Additionally, a key tactic for lowering the prevalence of diabetes and its effects is early and efficient diabetes screening. Despite a few flaws, the current book appears very good in this aspect. The article should, however, be released following a few significant revisions, like:
- The authors have focused on many invasive techniques for the detection of T2D. However, saliva was not explored as a non-invasive technique. In addition, the authors did not mention the promising diagnostic value of salivary glucose, its non-invasive nature, or its potential correlation with blood results.
- The authors have not discussed the implications of oxidative stress in diabetes. I suggest discussing reactive oxygen species and the degrees of glycation of antioxidant molecules like superoxide dismutase for the detection of diabetes.
a. A structural study on the protection of glycation of superoxide dismutase by thymoquinone. https://doi.org/10.1016/j.ijbiomac.2014.06.003.
- Mention the role of oxidative stress and advanced glycation endproducts in the detection of the severity of Provide the significance of oxidative stress markers in the early detection of T2D.
a. Therapeutic Potential of Myrrh, a Natural Resin, in Health Management through Modulation of Oxidative Stress, Inflammation, and Advanced Glycation End Products Formation Using In Vitro and In Silico Analysis. https://doi.org/10.3390/app12189175.
- HLA typing is an important diagnostic tool for the detection of several diseases, including T2D. I suggest highlighting the role of HLA typing using saliva as a significant diagnostic strategy for T2D.
- The authors did not provide the clinical data to support various strategies.
- The authors did not provide references to several biomarkers discussed in Table 1.
- Whole genome sequencing and next-generation sequencing could be a potential platform for the detection of several anomalies, including diabetes. In this regard, WGS and NGS should be discussed elaborately in separate subsections.
- I suggest changing the title of the manuscript. Biopsy does not look suitable for such good work. Instead of biopsy, the authors could use other scientific terms, such as innovative diagnostic strategies.
- The conclusions are poorly written compared to such a nice article. I think a conclusion should be written, highlighting the major findings of the review, limitations, and significance.
- Which method discussed in the manuscript is most affordable for the diagnosis of T2D?
- A table is suggested to compare the cost-effectiveness of traditional diagnostic tools and innovative liquid biopsy biomarkers for T2D detection.
- The novelty character of paper should be better marked.
- Careful attention needs to be paid to correct all the grammatical errors and reduce the repetition of the same explanations in different sections of the manuscript.
- Improve the writing of the objectives.
Comments on the Quality of English Language
Careful attention needs to be paid to correct all the grammatical errors.
Author Response
We thank the reviewer for the valuable feedback which helped improve our manuscript.
- The authors have focused on many invasive techniques for the detection of T2D. However, saliva was not explored as a non-invasive technique. In addition, the authors did not mention the promising diagnostic value of salivary glucose, its non-invasive nature, or its potential correlation with blood results.
A separate section regarding the use of saliva markers was included in the revised manuscript.
- The authors have not discussed the implications of oxidative stress in diabetes. I suggest discussing reactive oxygen species and the degrees of glycation of antioxidant molecules like superoxide dismutase for the detection of diabetes.
- A structural study on the protection of glycation of superoxide dismutase by thymoquinone. https://doi.org/10.1016/j.ijbiomac.2014.06.003.
We have discussed this in the revised version of the manuscript.
- Mention the role of oxidative stress and advanced glycation endproducts in the detection of the severity of Provide the significance of oxidative stress markers in the early detection of T2D.
- Therapeutic Potential of Myrrh, a Natural Resin, in Health Management through Modulation of Oxidative Stress, Inflammation, and Advanced Glycation End Products Formation Using In Vitroand In Silico https://doi.org/10.3390/app12189175.
We have discussed this in the revised version of the manuscript and added the suggested reference
- HLA typing is an important diagnostic tool for the detection of several diseases, including T2D. I suggest highlighting the role of HLA typing using saliva as a significant diagnostic strategy for T2D.
The role of HLA typing was discussed in the revised manuscript
- The authors did not provide the clinical data to support various strategies.
The clinical data was added for the different strategies used.
- The authors did not provide references to several biomarkers discussed in Table 1.
Table 1 was improved.
- Whole genome sequencing and next-generation sequencing could be a potential platform for the detection of several anomalies, including diabetes. In this regard, WGS and NGS should be discussed elaborately in separate subsections.
The use of NGS technical variants is included in the manuscript.
- I suggest changing the title of the manuscript. Biopsy does not look suitable for such good work. Instead of biopsy, the authors could use other scientific terms, such as innovative diagnostic strategies.
The manuscript title was changed, as suggested.
- The conclusions are poorly written compared to such a nice article. I think a conclusion should be written, highlighting the major findings of the review, limitations, and significance.
The conclusion and the challenges and perspective sections were revised.
- Which method discussed in the manuscript is most affordable for the diagnosis of T2D?
We have discussed the cost benefits of these methods in the conclusions section.
- A table is suggested to compare the cost-effectiveness of traditional diagnostic tools and innovative liquid biopsy biomarkers for T2D detection.
At the moment it is hard to envision the costs per patient for some of these analysis. However, we discuss the challenges regarding cost-effectiveness in the revised manuscript.
- The novelty character of paper should be better marked.
The novelty of the manuscript was highlighted and the objectives of the review clearly stated.
- Careful attention needs to be paid to correct all the grammatical errors and reduce the repetition of the same explanations in different sections of the manuscript.
This was checked and corrected.
- Improve the writing of the objectives.
The objectives were revised.
We have also improved Table 1 and added more references.
Reviewer 2 Report
Comments and Suggestions for Authors
Although the focus of this paper is interesting, it does not present enough data to qualify as a review.
1. Authors should indicate what keywords they used to search the literature in order to write this review.
2. Authors should address DNA and RNA related to pancreatic alpha cells, autoimmunity, SGLT, DPPIV, mitochondria, PPARs, sex hormones, racial differences, etc. in this review.
3. The table has no title and no table description.
4. The new biomarkers listed in the table are not specific and have insufficient cited papers.
5. There are insufficient cited papers related to the classical and inflammatory biomarkers listed in the table.
Author Response
We thank the reviewer for the valuable feedback which helped improve our manuscript.
- Authors should indicate what keywords they used to search the literature in order to write this review.
This paragraph was included in the revised manuscript.
- Authors should address DNA and RNA related to pancreatic alpha cells, autoimmunity, SGLT, DPPIV, mitochondria, PPARs, sex hormones, racial differences, etc. in this review.
We have addressed these markers in the revised version of the manuscript, as highligted in the new version of the manuscript.
- The table has no title and no table description.
The table was improved, a description and abbreviations were added
- The new biomarkers listed in the table are not specific and have insufficient cited papers.
Table 1 was significantly modified in order to address these issues.
- There are insufficient cited papers related to the classical and inflammatory biomarkers listed in the table.
New references were added in order to improve the markers listed in Table 1. New references are highlited in the revised manuscript.
Round 2
Reviewer 1 Report
Comments and Suggestions for Authors
I'm glad the challenges mentioned in the original manuscript have been fully resolved in the updated version, and I recommend that it now be accepted for publication.
Author Response
We thank the reviewer for their valuable feedback.
Reviewer 2 Report
Comments and Suggestions for Authors
The parts of the table other than "Novel Proposed Biomarkers" are widely known facts and do not need to be pointed out in this review. What is new in this review is the "Novel Proposed Biomarkers" section of the table. I think this review should be summarized as a mini-review focusing on the ”Novel Proposed Biomarkers”” section of the table.
The keywords below are not enough. "type 2 diabetes", "biomarker", "microRNA", "liquid biopsy" and "cell free DNA." In order to search for diabetes biomarkers, we believe that it is necessary to search the literature using the following keywords. Insulin resistance, insulin secretion, pancreatic beta cells, pancreatic alpha cells, obesity, pancreas, GLUT, DPPIV, SGLT, incretin, insulin, HNF, PPAR, and other molecules related to diabetes.
Author Response
Dear reviewer,
We understand your point of view regarding the new biomarkers. However, the scope of this review is the use of liquid biopsy for type 2 diabetes., a technique generally used for cancer patients but which has potential for use in other diseases as well. This is why we selected keywords relevant for liquid biopsy markers such as exosomes and cfDNA. The purpose of the review is not to discuss different diabetes biomarkers but to highlight the potential of liquid biopsy (from blood but also from saliva samples) to diagnose diabetes and its complications by using different biomarkers. These markers could potentially be included in point of care devices to further detect diabetes (and its complications) in a fast, cheap and non-invasive manner (for instance using saliva samples).
In table 1, we added the different classical biomarkers as requested in the first round of review. Regarding the novel biomarkers, these are not well investigated and need further validation. We have specified this in the revised version of the manuscript.
We thank you for your time in evaluating this manuscript.